# Is Bipolar Disorder the Consequence of a Genetic Weakness or Not Having Correctly Used a Potential Adaptive Condition?

**DOI:** 10.3390/brainsci13010016

**Published:** 2022-12-22

**Authors:** Mauro Giovanni Carta, Goce Kalcev, Alessandra Scano, Diego Primavera, Germano Orrù, Oye Gureye, Giulia Cossu, Antonio Egidio Nardi

**Affiliations:** 1Department of Medical Sciences and Public Health, University of Cagliari, Monserato Blocco I (CA), 09042 Cagliari, Italy; 2International Ph.D. in Innovation Sciences and Technology, University of Cagliari, Via Università 40, 09124 Cagliari, Italy; 3Department of Surgical Sciences, University of Cagliari, Asse Didattico Medicina P2—Monserrato (CA), 09042 Cagliari, Italy; 4WHO Collaborating Centre for Research and Training in Mental Health, Neuroscience and Substance Abuse, Department of Psychiatry, University of Ibadan, Oduduwa Road, Ibadan 200132, Nigeria; 5Laboratory Panic and Respiration, Institute of Psychiatry (Ipub), Federal University of Rio De Janeiro (Ufrj), Rio De Janeiro 22725, Brazil

**Keywords:** bipolar disorder, hyperactivity evolutionary perspective, RS1006737, CACNA1C

## Abstract

It is hypothesized that factors associated with bipolar disorder could, uer defined conditions, produce adaptive behaviors. The aim is to verify whether a genetic feature associated with bipolar disorder can be found in people without bipolar disorder but with hyperactivity/exploration traits. Healthy old adults (N = 40) recruited for a previous study on exercise were subdivided using a previously validated tool into those with and without hyperactivity/exploration traits and compared with a group of old patients with bipolar disorder (N = 21). The genetic variant RS1006737 of CACNA1C was analyzed using blood samples, DNA extraction, real-time PCR, FRET probes, and SANGER method sequencing. People with hyperactivity/exploration traits and without bipolar disorder were like people with bipolar disorder regarding the frequency of the genetic variant (OR = 0.79, CI95%: 0.21–2.95), but were different from people without either hyperactivity/exploration traits and bipolar disorder (OR = 4.75, CI95%: 1.19–18.91). The combined group of people with hyperactivity/exploration traits without bipolar disorder plus people with bipolar disorder had a higher frequency of the variant than people without either hyperactivity/exploration traits or bipolar disorder (OR = 4.25, CI95%: 1.24–14.4). To consider the genetic profile of bipolar disorder not an aberrant condition opens the way to a new approach in which the adaptive potential would be a central point in psychosocial treatment in addition to drug therapy. Future research can confirm the results of our study.

## 1. Introduction

It has been hypothesized that specific personality traits (or traits of temperament according to Akiskal’s definition) associated with a high risk of psychopathology, specifically a high risk of bipolar disorder, could, under defined conditions, produce adaptive behaviors [1,2,3]. This theory was supported by observations of how, in the face of rapid social change, people with aptitudes for exploration and hyperactivity could operate a culture “leap”, allowing them to acquire “new” and unusual behaviors for their culture of origin while “winning” in terms of new social and economic needs [4]. Such a revolutionary leap would not be without dangers; those who adapt will acquire relevant roles in the new context, while, in contrast, those who try the “leap” without success will be at high risk of psychopathology. This model could explain the epidemic of mood disorders that, according to some lines of research, would have developed in the modern era starting from the “English disease” of the early stages of industrialization [5,6,7].

Based on those observations, a study on migrants from Sardinia to Latin America has been developed. This study showed that migrants who had freely decided to leave for new destinations and second-generation sons of migrants with both Sardinian parents, living in megacities such as Buenos Aires or Sao Paulo, had more frequent episodes of sub-clinical hypomania than the Sardinians who had remained in Sardinia, even if they had an equal frequency of overt mood disorders [8]. This evolutionary/socio-biological vision of the bipolar spectrum could explain how the alleged basic vulnerability linked to a specific pattern of the response of social and biological rhythms [9] can, in contrast, represent a potential advantage in modern megacities, immersed in light and noise pollution, “in which life runs 24 h a day for seven days a week” [10].

The current work seeks to further explore this link. Specifically, it aims to verify whether a genetic feature associated with bipolar disorder [11] can be found in people without bipolar disorder but with characteristics of hyperactivity and exploration. Of the five genetic variables that have been found associated with bipolar disorder in the literature [11], the focus of the current exploration is the CACNA1C gene. Other susceptibility genetic variables associated with bipolar disorder are the ANK3, NCAN, ODZ4, SYNE1, and TRANK1 genes [11]. A previous study has led to the hypothesis that these gene genetic variables have a high frequency in elderly people who chose to participate in an active aging project (therefore with possible characteristics of expression and hyperactivity) [12]. However, this study did not have a valid and reliable tool that could define who had the supposed characteristics of exploration and hyperactivity, i.e., the criterion for defining the possession of these attitudes was very generic (having chosen to participate in an active aging project) [12].

## 2. Materials and Methods

### 2.1. Participants

The target population included elderly people living in an urban area recruited for a previous study on physical exercise [13,14]. A specifically validated ad hoc questionnaire was used to identify people with hyperactivity [15]. The following inclusion criteria were implemented: elderly people aged 60 or over of both genders, living at home, and having the capacity to provide informed consent. Additional inclusion criterion was the absence of a lifetime history of diagnosed bipolar spectrum disorder conditions. Exclusion criteria were: non-acceptance to participate in the study (not signing the informed consent for the study); participants with a diagnosis of bipolar spectrum disorder; and those under 65 years of age.

The control group involved patients with a formal diagnosis of bipolar spectrum disorders, aged 60 or over, of both genders, from the province of Cagliari. These subjects were identified through the registers of patients belonging to the Centro di Psichiatria di Consultazione e Psicosomatica (AOU Cagliari).

### 2.2. Psychiatric and Hyperactivity Evaluation

Both groups underwent psychiatric evaluation that included the collection of personal and family history. The measure of hyperactivity was carried out using the “Questionnaire for adaptive hyperactivity and goal achievement” (AHGA), a tool created and validated specifically for this purpose [15].

### 2.3. Genetic Procedure

Blood sampling of the 61 participants was performed at the Laboratory of Molecular Biology (Department of Surgical Sciences, University of Cagliari). Genomic DNA was extracted from the blood using the Bosphore Viral DNA/RNA Extraction Spin Kit (Anatolia Geneworks, Hasanpaşa, Turkey). Oligonucleotides for PCR (PRIMER) and fluorescent hybridization probes have been designed to detect CACNA1C polymorphisms. The application softwares used were Oligo version 6, Mfold, and Geneious. This molecular system involves the use of fluorescent hybridization probes (HP) that exploit the principle of FRET, or frequency resonance energy transfer [16]. The choice of these fluorescent probes is due to the fact that they are able to discriminate even the variation of a single nucleotide between the typical target sequences for bipolar disorder. At the end of the PCR, a constant speed heating of the Probe-DNA system is carried out, and there is a temperature where 50% of the probes detach from the target fragment, creating a lowering of the background fluorescence, the melting temperature (Tm). The system software shows this temperature via a spike in a graph that relates the negative-dt first derivative of the fluorescence to the Tm. The system displays the difference of a nucleotide along the DNA fragment by creating peaks in different locations. To confirm the specificity of the method, all samples were also analyzed by sequencing (Sanger method). The postanalytical phase consisted of the processing of data and the comparison, through application software, of the sequences obtained. The detection of the SNPs of interest was carried out using the programs Multiple Sequence Alignment Program Clustal Omega (ebi.ac.uk/Tools/msa/clustalo/ (accessed on 21 November 2022)) and Basic Local Alignment Search Tool (blast.ncbi.nlm.nih.gov/Blast.cgi (accessed on 21 November 2022)).

### 2.4. CACNA1C Gene and Risk Genetic Variant RS1006737

CACNA1C is a large gene located on the short arm of chromosome 12p13.3, with over 11,541 established variants. The majority of these variants are found in introns and downstream regions of the gene. CACNA1C encodes the alpha-1 subunit of a voltage-dependent calcium channel. This subunit creates a transmembrane channel that allows calcium ions to enter the cell [17]. Calcium channels are important neuronal regulators of heart muscle contraction, but they also play a role in skeletal muscle contraction. It is supposed that this gene is involved in axon guidance and synaptic transmission in the brain [18]. It appears that CACNA1C has a wider clinical genetic association with mental disorders than BD. Recent research indicates a link between the CACNA1C genotype and schizophrenia, major depressive disorder, and autism [17]. Missense mutations identified in this gene are responsible for an autosomal dominant genetic disorder named Timothy syndrome [19]. Moreover, the risk allele RS1006737 is recognized as the most cited and studied genetic risk for BD. Some researchers have hypothesized that this variant is able to influence gene expression, and others have indicated an effective variation of CACNA1C mRNA in post-mortem brain studies. CACNA1C’s rs1006737 mutant allele is associated with paranoid ideation, extraversion, trait anxiety, greater harm avoidance, and novelty seeking [20]. Moreover, it is found that CACNA1C rs1006737 is slightly associated with “flight of ideas” (disorganized thought) [20]. This clinical symptom is correlated with DSM-IV-classified manic episodes. It is assumed that this SNP may play a role in altering affective behaviors [20]. The link between CACNA1C risk variants and changes in brain volume is still being debated. According to some reports, this SNP is linked to increased subcortical volume, brainstem alterations, and increased grey matter density [20]. According to the study of Perrier et al. (2011), it was observed that carriers of this SNP have increased grey matter density in the right amygdala and the right hypothalamus [21]. A smaller left putamen was detected in BD patients carrying this risk allele in comparison to the healthy controls, suggesting that the rs1006737 polymorphism may impact anatomical variation within subcortical regions incorporated in emotional processing [21]. The frequency of this allele appears to be associated with the ethnic origin of the analyzed population, for example, about 54% of people of African descent and 5% of the Asian population share this allele [18]. FRET probes are able to distinguish the presence of both G and A alleles with specificity and selectivity errors (heterozygous profile).

### 2.5. Statistical Analysis

Chi-square values, P-values, odds ratios (ORs), and 95% confidence intervals (95% CI) were calculated for the genetic variant RS1006737 regarding its frequency in the three groups of participants: (a) with bipolar disorder; (b) with hyperactivity but without bipolar disorder; and (c) without hyperactivity and without bipolar disorder. The same statistical measures were also applied when comparing the genetic profile between combined groups of participants: the group of people with hyperactivity without bipolar disorder plus people with bipolar disorder, and the group of people without either hyperactivity or bipolar disorder.

### 2.6. Ethical Considerations

Approval for the study was granted by the Ethical Committee of the Institutional Review Board of the University Hospital of Cagliari, Italy (authorization signed on 11 July 2022, with a reference number: NP/2022/2893). All included subjects provided written informed consent.

The methodology presents some advantages:

(1) Working with older people makes it less likely that people identified as “not affected by bipolar disorder” will later develop the disorder. Considering that it is assumed that people with characteristics of hyperactivity may be at high risk for bipolar disorder.

(2) Older adults who join an active aging program could more likely have characteristics of hyperactivity, thus facilitating the recruitment of people without lifetime mood disorders but with features of hyperactivity.

(3) The fact that the recruited individuals are all from urban areas increases, according to our hypothesis, the probability of a higher frequency of people with hyperactivity.

## 3. Results

Table 1 shows the characteristics of the sample evaluated by gender in the three groups of people: (a) with bipolar disorder, (b) with hyperactivity without bipolar disorder, and (c) without hyperactivity and without bipolar disorder. All participants are 60 or older, of both genders. Table 2 shows that people with hyperactivity and without bipolar disorder (+H-BP) are homogeneous with people with bipolar disorder (+BP) as regard the frequency of the genetic variant RS1006737 (OR = 0.79, CI 95% 0.21–2.95), but not with people without hyperactivity and without bipolar disorder (-H-BP) (OR = 4.75, CI95% 1.19–18.91). If the group with hyperactivity and without bipolar disorder is added to the group with bipolar disorder (Table 3), the set of the two groups has a frequency of the variant RS1006737 that is clearly higher than that of the group without hyperactivity and without bipolar disorder (OR = 4.25, CI95% 1.24–14.4).

## 4. Discussion

The results of this study found that a genetic variant, RS1006737, recognized in the literature as associated with bipolar disorder [20,21], was found to have a high frequency (76% of individuals) in people without bipolar disorder but with traits of hyperactivity; with a comparable frequency to those of individuals with bipolar disorder (71%), but with a much higher frequency to individuals without hyperactivity traits and without bipolar disorder (40% of individuals). The positive subjects of the genetic variant RS1006737 identified in our study were all heterozygous (GA). RS1006737 as a single-nucleotide polymorphism is considered a functional polymorphism because its homozygous (AA) genotype has been linked with higher CACNA1C messenger RNA expression in the prefrontal cortex (PFC) compared with non-carrier (GG) or heterozygous (GA) genotypes [22]. Moreover, it has been reported that BD carriers of the A allele have elevated amygdala activity during emotional processing tasks in contrast with non-carriers [22]. A recent study described that BD subjects carrying the CACNA1C risk allele A had elevated levels of intracellular calcium in comparison with healthy controls [23].

Several factors need to be highlighted in considering the results of this study. First, the sample of older adults with hyperactivity without bipolar disorder may have been characterized by a more active social network given that they were recruited through media methods or the territorial health network; second, the individuals in this group were without lifetime mood or other mental health disorders; third, they were in relatively good health or only had ailments that were not uncommon among the elderly (hypertension, diabetes, etc.) and which did not compromise their ability to carry out mild to moderate physical activity; and fourth, they were sufficiently enterprising to start a demanding program of physical exercise. Taking these observations into consideration, our results seem to confirm the hypothesis that some basic characteristics that are typical of bipolar disorder (including genetic characteristics) are not always associated with an increased risk of disease but could also have an adaptive significance in certain circumstances. The interactions between genetic heritage and environmental factors have aroused specific interest in the neurogenesis of the Hypothalamic-Pituitary-Adrenal (HPA) axis and in its consequences for emotional and behavioral responses [24,25,26]. It is known that HPA axis activation by social and psychological stress could increase the risk of the onset of mental conditions [27,28]. Specifically, bipolar disorder was found “associated with dysfunction of HPA axis activity” [29,30].

Chronic severe stress leads to changes in the functioning of the prefrontal and limbic systems with influence on the regulation of gene expression and neuroendocrine, emotional, and behavioral responses [26,28,31]. The change/increase in rhythms of modern and urban life could influence the HPA axis in adults and older adults [27,32,33]. Inhabitants of cities can access better resources than those living in rural areas, such as education, leisure and cultural activities, and economic and healthcare opportunities [24,32]. However, surprisingly, mood disorders show higher prevalence in urban areas [34,35,36,37], and a dose-dependent association has been found between exposure to the urban environment and the onset of severe mental health episodes as well as a negative prognosis [38]. Therefore, living in cities is associated with more frequent achievement of social goals but also with a higher risk of impaired mental health, including mood disorders. Our group formulated some hypotheses about how cities’ noise and light pollution might influence mental health and increase the risk for bipolar disorder [10].

People with novelty-seeking aptitudes and explorers/and “challengers” with hyper-rhythmic temperaments/personalities could have adaptative resources in a new environment, as confirmed in the above-cited studies on voluntary migrants (not refugees) [8]. In fact, if modern life demands that biological rhythms be broken, it could favor people with a basic predisposition to changing and adapting to biological and social rhythms in different ways, i.e., people who can mobilize their energy for some (even limited) periods and with less need for sleep [10]. The supposed increase in mood disorders would be the pathological side, resulting from the failure of the demands for adaptation, such as when the challenges outweighed the ability to adapt or when the increase in energy and the reduced need for sleep have not been sufficient to achieve the desired goals. Thus, the result can be an adaptation at some times, but at other times, when the demand is too stressful, adaptation fails, and a “new pathology” emerges with features of broken biological/social rhythms.

The “English Illness”, that is, the mood disorders with new psychopathological characteristics, was born, according to Murphy, in the newly industrialized areas [5]. Supporting this hypothesis, several surveys have found a continuous increase of mood and bipolar disorders in Western societies [6,7,39]. Artificial light induces, during natural darkness, activities usually performed in daylight, such as work activities, food intake, or social life [10,40]. This impacts the immune-endocrine and other biologic rhythms, which evolution finalized to synchronize human behavior with variations in daylight and changes in seasons [41]. It was found that the risk of bipolar increased with disorders of the sleep-wake cycle and with artificial light pollution [42]. Melatonin decreases estradiol and increases progesterone levels [43], and light pollution may lower melatonin secretion, changing the estradiol/progesterone ratio. Thus, the increase of estradiol and other “stimulating” steroid hormones induced by light pollution may have a role in the increasing risk of bipolar disorder [27,44].

If our hypothesis about the evolutionary genesis of bipolar disorder is confirmed, this will not change much about the pharmacological approach. In other words, if an imbalance in adapting to biological rhythms as we have postulated is central, a mood stabilizer would still be the most appropriate tool based on current knowledge. The socio-biological perspective would, however, change the approach to what until now has been defined as “rehabilitation” and “psycho-social approach”. The new interpretation would, in fact, open the way to a new approach to supporting drug therapy in which the rediscovery of the adaptive potential would be central for the individual who has suffered a decompensation. Furthermore, the interpretation of the disorder as more than simply the consequence of a genetic weakness would be an element against stigma and self-stigma.

Some evident limits of our study need to be underlined.

First of all, the study is exploratory since it is based on a small sample, and its results need to be confirmed with a much larger sample. The sample of old adults without bipolar disorder shows a high frequency of people with hyperactivity (25 out of 40 = 62.5%); the modality of recruitment of the sample (old adults who participated in an active aging project living in an urban area) may have favored the selection of people with these features. This is not necessarily a cause for bias because this was intended to empower the power of the study. Future studies will need to establish the true frequency of hyperactivity in the general population. However, the concomitantly high percentage of people with the gene variant found in our sample (much higher than expected in the general population based on previous studies [20,21]) confirms that this is a non-representative sample of the general population and that hyperactivity is overrepresented. This relatively undermines the strength of the association between hyperactivity and the presence of the variant and does not affect the value of the results.

## 5. Conclusions

This study found that a genetic variant, RS1006737, recognized in the literature as associated with bipolar disorder, was found in well-adapted older adults without bipolar disorders and high hyperactivity traits with a similar frequency as in older adults with a diagnosis of bipolar disorder and higher than in older adults without bipolar disorder and without hyperactivity. The study involved a very small sample, and its results need to be confirmed. If the results and the hypothesis of an evolutive genesis of bipolar disorders are confirmed, the new interpretation could open the way to a new approach to supporting drug therapy in which the rediscovery of the adaptive potential resources would be central to the recovery of the individual who has suffered a bipolar disorder onset. Furthermore, the interpretation of the disorder as not simply the consequence of a genetic weakness could be an element against stigma and self-stigma.

## Figures and Tables

**Table 1 brainsci-13-00016-t001:** Characteristics of the sample evaluated by gender in the three groups of people: (a) with bipolar disorder, (b) with hyperactivity without bipolar disorder, and (c) without hyperactivity and without bipolar disorder.

		People with Bipolard Disorder	People with Hyperactivity, but without Bipolar Disorder	People without Hyperactivity, and without Bipolar Disorder
		N = 21	N = 25	N = 15
**Gender**	Man	7 (33.4%)	16 (64%)	6 (40%)
	Woman	14 (66.6%)	9 (36%)	9 (60%)

**Table 2 brainsci-13-00016-t002:** People with hyperactivity and without bipolar disorder (+H-BP) are homogeneous as regards the frequency of variant RS1006737 in the people with bipolar disorder (+BP) but not the people without hyperactivity and without bipolar disorder (-H-BP).

Groups	Gen+ Variant (RS1006737)	Gen- Variant (RS1006737)	Total	Homogeneity with +H-BP	P	OR +H-BP	CI 95%
People with bipolar disorder	15 (71%)	6	21	χ^2^ = 0.124	0.725	0.79	0.21–2.95
People with hyperactivity, but without bipolar disorder	19 (76%)	6	25	Pivot			
People without hyperactivity, and without bipolar disorder	6 (40%)	9	15	χ^2^ = 5.184	0.023	4.75	1.19–18.91

**Table 3 brainsci-13-00016-t003:** Difference in genetic profile (variant RS1006737) in people with hyperactivity (with or without BD) plus bipolar disorders and people without hyperactivity and without bipolar disorder.

Groups	Gen+ Variant (RS1006737)	Gen- Variant (RS1006737)	χ^2^	P	OR	CI 95%
People with hyperactivity, but without bipolar disorder + people with bipolar disorder	34 (74%)	12	5.763	0.016	4.25	1.24–14.4
People without hyperactivity and without bipolar disorder	6 (40%)	9				

## Data Availability

The data presented in this study are available on reasonable request from the corresponding author. The data are not publicly available due to privacy restrictions.

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
