# Peer review of "Is Bipolar Disorder the Consequence of a Genetic Weakness or Not Having Correctly Used a Potential Adaptive Condition?"

_brainsci, 2022, doi:10.3390/brainsci13010016_

Round 1

Reviewer 1 Report

Dear Authors

This is a very interesting article on bipolar disorder. However, you have to pay attention at some points.

· do not use acronyms in the abstract

· “The methodology presents some advantages……..” should be moved to the methodology section?

· explain the meanings of the variables under consideration

· By which method the statistical analysis was done?

· The results section is too small. A table with the demographics is missing and why not a figure with the most important results?

· You provide a fairly long discussion without being supported by corresponding results.

Author Response

Cagliari, Italy, December 5th, 2022

Re: Manuscript ID brainsci-2059258

Response to Reviewers of the manuscript entitled “Is bipolar disorder the consequence of a genetic weakness or not having correctly used a potential adaptive condition”

Dear Reviewer,

Thank you so much for your extensive, insightful comments. Please find below a point-to-point response to your remarks. The corresponding editing in the text has been highlighted in green for your convenience.

Referee 1.

Q1: Do not use acronyms in the abstract.

R1: Thanks for the comment. All acronyms in the abstract have been removed.

Q2: “The methodology presents some advantages……..” should be moved to the methodology section?

R2: Thanks for the comment. The suggested part was moved to the materials and methods section.

Q3: Explain the meanings of the variables under consideration.

R3: Thanks for the comment. The significance and role of the genetic variant involved in our study is fully explained in the subsection ‘‘CACNA1C gene and risk genetic variant RS1006737’’, part of the section materials and methods.

Q4: By which method the statistical analysis was done?

R4: The subsection statistical analysis is added to the section materials and methods.

Q5: The results section is too small. A table with the demographics is missing and why not a figure with the most important results?

R5: Thanks for the comment. A table with the demographics regarding gender of all participants is included in the section Results, as Table 1.

Q6: You provide a fairly long discussion without being supported by corresponding results.

R6: Thanks for the comment. The discussion part apart from the tables is also concentrated on the evolutionary perspective of bipolar disorders which underlines and supports the results. A new part about the RS1006737 as a functional single-nucleotide polymorphism present in homozygous or heterozygous form, or not present at all is added.

        Dr. Alessndra Scano, Ph.D on behalf of the co-authors

Reviewer 2 Report

My suggestions:

1. I would add a brief introduction to the CACNA1C gene. For example, what is its function, its structure, and what diseases could be involved in?

2. Were there any other genetic risk factors described for bipolar diseases before? Authors may mention it either in the introduction or in the discussion. 

3. There are younger patients too, diagnosed with BD. Why they were not included in the study? Later, it would be interesting to compare the genetic pattern of younger and older patients with BD. In Methods, a table may be needed, which introduces the demographic pattern of patients with BD and the controls. 

4. Does it impact the BD symptoms or presence, whether RS1006737 is present in homo-or heterozygous form? 

5. RS1006737 is an intronic variant. It would be interesting to mention, how it could impact CACNA1C function, leading to BD/hyperactivity.

Author Response

Cagliari, Italy, December 5th, 2022

Re: Manuscript ID brainsci-2059258

Response to Reviewers of the manuscript entitled “Is bipolar disorder the consequence of a genetic weakness or not having correctly used a potential adaptive condition”

Dear Reviewer,

Thank you so much for your extensive, insightful comments. Please find below a point-to-point response to your remarks. The corresponding editing in the text has been highlighted in green for your convenience.

 Referee 2.

 Q1: I would add a brief introduction to the CACNA1C gene. For example, what is its function, its structure, and what diseases could be involved in?

R1: Thanks for the comment. The structure, function and what diseases could be involved in are fully explained in the subsection ‘‘CACNA1C gene and risk genetic variant RS1006737’’, part of the section Materials and Methods.

Q2: Were there any other genetic risk factors described for bipolar diseases before? Authors may mention it either in the introduction or in the discussion.

R2: Thanks for the comment. Other genetic risk factors are mentioned in the introduction section.

Q3: There are younger patients too, diagnosed with BD. Why they were not included in the study? Later, it would be interesting to compare the genetic pattern of younger and older patients with BD. In Methods, a table may be needed, which introduces the demographic pattern of patients with BD and the controls.

R3: Thank you for your relevant input. Indeed, our next phase is including the younger patients with BD too. Regarding the table that introduces the demographic pattern of patients with BD and the controls, it is placed in the Results section as Table 1.

Q4: Does it impact the BD symptoms or presence, whether RS1006737 is present in homo-or heterozygous form?

R4: Thanks for your comment. The impact of the RS1006737 single nucleotide polymorphism in homozygous, heterozygous form, or in non-carrier form at all is explained in more detail in section discussion.

Q5: RS1006737 is an intronic variant. It would be interesting to mention, how it could impact CACNA1C function, leading to BD/hyperactivity.

R5: Thanks for your comment. The impact and association of the genetic variant RS1006737 with BD and the traits of hyperactivity is explained in the subsection ‘‘CACNA1C gene and risk genetic variant RS1006737’’, part of the section materials and methods.

Dr. Alessndra Scano, Ph.D on behalf of the co-authors

Round 2

Reviewer 2 Report

The authors fulfilled my suggestions, thank you.